# Implementation of FAIR Principles for Ontologies in the Disaster Domain: A Systematic Literature Review

**Allan Mazimwe** [1,*,†]**, Imed Hammouda** [2,†] **and Anthony Gidudu** [1,†]

1   Department of Geomatics and Land Management, Makerere University, Kampala, Uganda;
    anthony.gidudu@mak.ac.ug
2   Mediterranean Institute of Technology, South Mediterranean University, Tunis 1053, Tunisia;
    imed.hammouda@medtech.tn
*   Correspondence: allanmazimwe@mak.ac.ug
†   These authors contributed equally to this work.

**Abstract:** The success of disaster management efforts demands meaningful integration of data that is geographically dispersed and owned by stakeholders in various sectors. However, the difficulty in finding, accessing and reusing interoperable vocabularies to organise disaster management data creates a challenge for collaboration among stakeholders in the disaster management cycle on data integration tasks. Thus the need to implement FAIR principles that describe the desired features ontologies should possess to maximize sharing and reuse by humans and machines. In this review, we explore the extent to which sharing and reuse of disaster management knowledge in the domain is inline with FAIR recommendations. We achieve this through a systematic search and review of publications in the disaster management domain based on a predefined inclusion and exclusion criteria. We then extract social-technical features in selected studies and evaluate retrieved ontologies against the FAIR maturity model for semantic artefacts. Results reveal that low numbers of ontologies representing disaster management knowledge are resolvable via URIs. Moreover, 90.9% of URIs to the downloadable disaster management ontology artefacts do not conform to the principle of uniqueness and persistence. Also, only 1.4% of all retrieved ontologies are published in semantic repositories and 84.1% are not published at all because there are no repositories dedicated to archiving disaster domain knowledge. Therefore, there exists a very low level of Findability (1.8%) or Accessibility (5.8%), while Interoperability and Reusability are moderate (49.1% and 30.2 % respectively). The low adherence of disaster vocabularies to FAIR Principles poses a challenge to disaster data integration tasks because of the limited ability to reuse previous knowledge during disaster management phases. By using FAIR indicators to evaluate the maturity in sharing, discovery and integration of disaster management ontologies, we reveal potential research opportunities for managing reusable and evolving knowledge in the disaster community.

**Keywords:** FAIR; ontologies; disaster management; interoperability

## 1. Introduction

The world is increasingly exposed to hazards due to climate change impacts [1]. These hazards affect the most vulnerable communities resulting in disasters. The success of disaster management efforts demands meaningful integration of both spatial and non-spatial data from various stakeholders. However, there exist semantic interoperability barriers in the data making it difficult to have meaningful data integration. This stifles the data integration process and stakeholder involvement during disaster management.

Ontologies help in organising both geospatial and non-spatial data within the disaster management domain [2] to guarantee semantic interoperability. Vast work exists on the development of ontologies for representing disaster management knowledge [3,4] though non expressively captures the entire scope of the domain. These ontologies remain underused in practice due to several reasons which include challenges in knowledge

discovery, the need to update disaster knowledge as the domain evolves [5–7], and lack of coordinated approach towards semantics [8].

Modern data management demands full support for FAIR (Findable, Accessible, Interoperable, Reusable) principles [9]. Wherefrom, the concept of Interoperability is defined recursively [10] such that data interoperability is achievable when corresponding semantic artefacts are FAIR [11]. Therefore, implementation of the FAIR principles for ontologies can help to solve the demands of data interoperability. This is because FAIR principles enable the discovery of interoperable ontologies that can handle heterogeneous and diverse information in the Disaster Management Domain. While there are initiatives for developing metrics to measure the adherence of ontologies to FAIR principles in multiple domains [12,13], little guidance exists on how to evaluate them. Furthermore, a major issue identified by [14] is that comparison of older versions of ontologies to understand their evolution is challenging. This is because these ontologies are not shared in line with best practices for semantic artefacts. Therefore, ability to share and discover knowledge will guarantee the reuse of evolving disaster management knowledge. However, limited literature [3,15] exists on the identification of gaps in how well disaster management science has been able to share knowledge and outcomes.

Therefore, the contribution of this paper is; First, we identify existing disaster management ontologies and associated features that affect the maturity for disaster knowledge discovery and sharing. Secondly, we provide a semi-quantitative understanding of the extent to which existing ontologies in the disaster domain are implemented in line with the FAIR agenda. Finally, we reveal research opportunities for managing evolving disaster knowledge. The rest of the paper is structured as follows. Related work is presented in Section 2. Materials and methods are introduced in Section 3 while results and discussions are given in Sections 4 and 5 respectively. Section 6 presents the limitations of the study. Finally, Section 7 details the main conclusions and recommendations.

## 2. Related Work

### 2.1. FAIR Principles and Disaster Data Management

Data in the disaster community are typically dispersed geographically and owned by various stakeholders rendering it inaccessible for reuse in research and policy formulation. FAIR principles proposed by [9] provide us with guidance on how to make digital resources Findable, Accessible, Interoperable and Reusable. While the concept of FAIR started with data [9,11], it has since been expanded to capture other digital resources such as Software [16], semantic artefacts [17], etc. In the Disaster Community, there exist explicit use cases applying FAIR principles to ensure that data are reusable. For instance, there exists an initiative by the Task Group on FAIR Data for Disaster Risk Research [18] to address challenges and best practices of disaster data management. Also, authors in [19] present work by the Koninklijk Nederlands Meteorologisch (KNMI) Instituut)/ORFEUS(ORFEUS-Observatories and Research Facilities for European Seismology) under European EUDAT project to share insight into climate change impacts and earthquakes (see https://tinyurl.com/BsafeSeismicdata Accessed on 2 December 2020) respectively using the BSAFE service. However, existing works in the disaster domain focus on the implementation of FAIR recommendations rather than establishing the maturity of disaster digital artefacts.

In the FAIR ecosystem, novel literature exists on FAIR metrics and evaluation criteria for digital objects [8,11,16]. For comparability of assessment approaches, there are ongoing initiatives to develop a FAIR maturity model [17,20] that combines various metrics into a generic self-assessment model for measuring the maturity level of a dataset. The maturity model proposes two perspectives on methodologies for evaluation [20]. The first perspective (i.e., progress measurement) measures the extent to which the evaluated resource meets the requirements expressed in an indicator. This indicates the steps for fully achieving the satisfaction of the indicator. The second perspective measures adherence of the evaluated resource to the requirements of an indicator on a binary Pass or Fail scale. This is used to understand how a resource performs with respect to a FAIRness target level.

A shortfall of these maturity models is that evaluation priorities may be different depending on the context. This may require upgrading and downgrading of indicators which has implications on evaluation methods and results obtained. Moreover, FAIR practices are likely to vary between communities due to a variety of requirements [8]. More generically, existing work on maturity models describes desired features that digital resources should posses to render them FAIR but provides little guidance to achieve these goals. Based on this observation, the authors in [21] have developed an example of what FAIR should look like and have proposed tooling to enhance discovery, integration and reuse of digital resources. In a different context, authors in [22] propose optimisation patterns for processes in the emergency response subdomain. The application of this work could guide data management processes to achieve FAIR goals in the disaster management domain.

*2.2. FAIR Ontologies in the Disaster Domain*

Existing FAIR initiatives in the disaster management domain [19] mainly focus on technical aspects for sharing data rather than semantic aspects. However, as explained in [10,17] data are FAIR when the schema/ontologies organising the data are themselves FAIR. Vast literature on the use of vocabularies (such as glossaries, thesauri and ontologies, etc) to organise data exists in the disaster domain [2]. Depending on the main focus of the agency/stakeholder, the concepts defined within these vocabularies vary. Best practices for publishing and reusing such ontologies have long been developed by the semantic community [13,23–26]. Bonatti et al. [27] have classified these best practice indicators for knowledge graphs into two categories i.e., its provision (driving "findable" and "accessible") and its design (driving "interoperable" and "re-usable"). Liu et al. [3] review ontologies, vocabularies and taxonomies in the disaster domain with a specific focus on crisis management in terms of their coverage, design and usability. Furthermore, Le Franc et al. [17] have recommended blueprint metrics and best practices for semantic artefacts. Such best practices include use of patterns proposed by Gangemi [28]. In the disaster management domain, a few such patterns are presented by [29,30]. However, an exhaustive set of best practices or standards may be non existent in the semantic community. The authors in [4] review ontology artefacts and their characteristics to motivate the development of the Empathi ontology in the emergency management. However, these efforts do not cover the full spectrum of FAIR principles. In addition, the ontologies mentioned cover the crisis management subdomain which is rather a subset of the disaster management domain. To the best of our knowledge, no work exists on a holistic evaluation of FAIRness for ontologies in the disaster domain.

## 3. Materials and Methods

In this study, we perform a Systematic Literature Review (SLR) based on PRISMA guidelines by [31] to evaluate the use of FAIR principles for publishing ontology artefacts in the disaster management domain. As such, we scope the SLR to a meta-analysis of formal ontologies developed in the disaster domain. To describe the disaster domain, we use the disaster management cycle components (Hazard, Vulnerability, Risk Analysis, Prevention and Mitigation, Preparedness and Early Warning, Disaster Response (humanitarian/relief and emergency response), Recovery and Reconstruction). Therefore, this study answers the following Research Questions (RQ);

- RQ1: *Which formal ontologies exist in the disaster domain?* We extract existing ontologies in the disaster management domain and their socio-technical features (i.e., composition of contributors in development teams, motivation and development approaches). Understanding socio-technical features associated with the ontologies is key to understanding the implementation of best FAIR recommendations.
- RQ2: *To what extent are formal ontologies in the disaster domain Findable, Accessible, Interoperable and Reusable (FAIR)?*

A maturity metric for data interoperability is that it should be organised using FAIR vocabularies. Therefore, assessing the extent to which disaster domain ontologies are inline with FAIR recommendations allows domain experts to understand the current landscape in the utilization of semantic best practices. This will enable the identification of gaps and solutions to improve data interoperability.

To answer research questions above, we systematically review ontologies for organising concepts along the disaster management cycle in literature and existing repositories.

Therefore, a search process was carried out through a combination of keyword and backward snowball search [32] of articles published in indexed electronic databases for the period between January 2010 to December 2019. The search was carried out on IEEE, ACM digital library, Google Scholar, ISI Web of science and Elseviers Scopus to ensure a comprehensive coverage of electronic databases. Using the search terms proposed in Table 1, the keyword-based searches retrieve papers of interest from the electronic databases by matching search words with the article title. Related keywords within each set are linked using the logical OR operator. Furthermore, the two sets are combined with the logical AND i.e.,

*(Ontology OR ODP OR Vocabulary OR Terminology)) AND*
*(Hazard OR Vulnerability OR Disaster OR Risk OR Crisis OR*
*Humanitarian OR Emergency OR "Early Warning")*

**Table 1.** Keyword Sets Ontologies in the Disaster Management Domain.

| keyword set 1: Ontologies |
| --- |
| Ontology |
| ODP (ontology design pattern) |
| Vocabulary |
| Terminology |
| **keyword set 2: Disaster management domain** |
| Hazard |
| Disaster |
| Vulnerability |
| Risk |
| Crisis |
| Humanitarian |
| Early Warning |
| Emergency |

The result set retrieved from the intersection of the two keyword sets in multiple databases was the starting point for a manual review of resources. We downloaded the retrieved studies into a spreadsheet to remove duplicate entries and perform further screening. A key issue in this screening is the choice of the search keywords. Articles that do not use the specified keywords in the title will not appear in the result set even when they have a strong thematic linkage to the topic of interest. Table 1 shows the search terms used to review existing ontologies in the domain.

To extract articles for the review, we performed a two-step screening process, i.e., preliminary and detailed screening. Preliminarily, we screened all abstract and keywords in retrieved articles to extract relevant studies. The full text of the resulting relevant studies were screened in detail to exclude those not eligible for inclusion in the review. The exclusion and inclusion criteria in the review protocol (see Sections 3.1 and 3.2) formed the basis for the screening process. To identify additional articles to include, a backward snowballing iteration [32] of reference lists for selected full-text studies was performed in line with inclusion and exclusion criteria.

### 3.1. Inclusion Criteria

The following rules were used to screen articles for the review. An article was selected if;

- Keywords from both word sets in Table 1 appear in the title. This criterion is applied to only articles obtained from database search only.
- Study carried out between 2010–2019 and is written in English language. This criteria is applicable to all articles extracted from database search and snowball sampling of reference lists in searched articles.
- The primary focus is the disaster management domain in the context of the natural environment (i.e., DRR, Disaster Emergency Management, etc.). This rule was applied to all articles at all stages of screening.
- It reports the development of a formal ontology artefacts for representing disaster domain concepts.

### 3.2. Exclusion Criteria

Furthermore, the following rules were used to exclude articles from the review during screening. An article was excluded if;

- It had corresponding duplicate paper(s) such that the two papers publish the same study. In such a case the less mature one is excluded in favour of the extended one. This was necessary to ensure same data are not counted twice.
- The content is not about hazards and disasters that occur in the context of the natural environment. For instance, all work on disasters in the context of computer software and information systems was excluded.
- The article full text is not accessible for detailed screening.
- It presented existing ontologies with no development of new ontology artefact being reported. An example of such articles includes review articles.

To answer RQ1, we perform a manual review of all selected publications and subsequently classify publications according to phases of the disaster management cycle. These include; Risk Assessment (Hazard, Vulnerability and Risk Analysis), Prevention and Mitigation, Preparedness and Early Warning, Response, and Recovery. To determine the social technical characteristics, we also extract motivations for developing new ontologies, development team composition and development approach from each of the selected articles through a manual review. To understand the development team expert composition, we searched publication author affiliations and scholar profiles on google.

In RQ2, ontologies in selected articles (see *Data Availability Statement*) are manually reviewed to extract ontology discovery, integration and reuse features in line with FAIR maturity recommendation (PRec and BPRec) in Table 2 adapted from [17]. Additional metrics (KRec) from [25] are adopted to improve the characterisation of reusability metrics. However, we exclude some metrics since it is not possible to evaluate them for extracted ontologies.

To measure FAIRness, we use the binary Pass-or-Fail scale [20] together with the modified indicator function from [33] as in Equation (1).

$$1_E(V) = \left\{ \begin{matrix} 0 \; if \; V \; \not\epsilon E \\ 1 \; if \; V \epsilon E \end{matrix} \right\} \tag{1}$$

In Equation (1), $V$ is the ontology being evaluated against an indicator $E$ in any FAIR component. We then represent the count of all ontologies that satisfy the function in Equation (1) for each indicator and compute it as a percentage of the total number of ontologies ($N_v$) in Equation (2).

$$RecE = \sum \frac{1_E(V) * 100}{N_v} \tag{2}$$

**Table 2.** FAIR Metrics for Ontologies, Adapted from [17,25].

| Principle | FAIR Metric (Indicator) |
|---|---|
| Findability | PRec.1: Use Globally Unique, Persistent and Resolvable Identifier for Semantic Artefacts and their content |
| | PRec.2: Use Globally Unique, Persistent and Resolvable Identifier for Semantic Artefact Metadata Record |
| | PRec.3: Use a common minimum metadata schema to describe semantic artefacts and their content |
| | PRec.4: Publish the Semantic Artefact and its content in a semantic repository |
| Accessibility | PRec.5: Semantic repositories should offer a common API to access |
| | PRec.7: Repositories should offer a secure protocol and user access control functionalities |
| | PRec.8: Define human and machine-readable persistency policies for semantic artefacts metadata |
| interoperability | PRec.9: Semantic artefacts should be compliant with Semantic Web and Linked Data standards |
| | PRec.10: Use a Foundational Ontology to align semantic artefacts |
| | PRec.12: Semantic mappings should use machine-readable formats based on W3C standards (R) |
| | PRec.13: Crosswalks, mappings and bridging between semantic artefacts should be documented, published and curated |
| | PRec.14: Use standard vocabularies to describe semantic artefacts |
| | PRec.15: Make the references to the reused third-party semantic artefacts explicit |
| Reuse | PRec.16: The semantic artefact should be clearly licensed for machines and humans |
| | PRec.17: Provenance should be clear for both humans and machines |
| | KRec.1: Competency questions (CQs) are specified (knowledge characterised by CQs supports reuse) |
| | KRec.2: Ontology axioms are available (a set of human-readable logical expressions presents explicit meanings) |
| | KRec.3: Schema Diagrams are provided (diagramatic expression of main ideas in the ontology facilitates reuse) |
| | KRec.4: Ontology artefacts are annotated |
| FAIR Best Practices | BPRec.3: Use defined ontology design patterns |
| | BPRec.8: Provide a structured definition for each concept(ontology classes are annotated) |

However, some community based recommendations may not be applicable for an individual ontology. In this situation, such indicators were eliminated from the analysis. From this equation, we define the value of each FAIR component i.e., *Findability(V)*, *Accessibility(V)*, *Interoperability(V)*, *Reuse(V)* as weighted average as shown in Equation (3).

$$FAIRComponent\_value = \sum \frac{w_i(RecE_i)}{n} \tag{3}$$

Each FAIR metric often has a weight (i.e., essential, important and useful). In this study, we assume all indicators are equally important since no basis for weighting exists in the semantic community. As such, the weight was considered as 1. Such that *n* is the number of indicators involved. In this study, we classify FAIR values using a 5 point grade scale (see Table 3).

**Table 3.** FAIR Classification Scale.

| Range | Class |
|---|---|
| 0 to 20% | very low |
| 20 to 40% | low |
| 40 to 60% | medium |
| 60 to 80% | high |
| 80 to 100% | very high |

We also searched several repositories (e.g., Github http://github.com/, accessed on 21 December 2020 and Semantic repositories within the Bartoc registry https://bartoc.org/registries, accessed on 21 December 2020) to validate the findability of retrieved ontologies and identify more disaster domain ontologies. To further provide insight into reuse, we created a word/tag cloud visualization of ontologies reused in constructing disaster management ontology artefacts. The word/tag cloud shows ontologies in varying sizes depending on the frequency of reuse. Large words in the tag cloud correspond to a higher frequency of reuse.

### 3.3. Threats to Validity

*Construct validity*: A threat to construct validity is the inadequate extraction of articles from electronic databases due to the choice of the keywords used in article search. Articles that do not use the specified keywords in title, abstract or keywords, will not appear in the result set even though they may have a strong thematic linkage to ontologies and disaster management. To anticipate missing papers during the automatic search, we performed pilot searches to tune the search criteria with a diverse list of keywords covering topics of interest. We triangulate keyword search on 5 electronic databases and further scan references for all retrieved studies to ensure wide coverage and representation.

*Internal validity*: To mitigate personal bias that can affect internal validity, three researchers independently performed data extraction and synthesis, wherefrom existing conflicts were discussed and resolved to extract at the selected studies.

*External validity*: This SLR follows a predefined protocol from [31] which is reproducible.

*Reliability*: To ensure that relevant studies were not excluded thus mitigating the threats to reliability, the selection process and the inclusion and exclusion criteria were iteratively designed and discussed by the authors to minimize the risk of exclusion of relevant studies.

### 4. Results

In Figure 1, an initial search yields 1593 articles from 5 electronic databases and 27 from additional sources. Furthermore, there is an exclusion of 320 duplicates from the total 1620 articles. Upon screening the remaining 1283 records, we eliminate 1119 records irrelevant to the search resulting in 162 eligible for review. The high number (1119 articles) of elimination is due to the vast ontology literature on disaster risk and vulnerability for computer and software systems that are not a focus of our review. Similarly, there exist philosophical articles defining ontology and epistemology in the disaster management domain. An assessment of the 162 records using predefined inclusion and exclusion criteria resulted in the exclusion of 93 articles while yielding 69 publications for qualitative and quantitative analysis.

The review presents results from journals, conferences and technical reports. This is acceptable with the exception of computer science [34] where peer-reviewed conference articles are also considered credible as journal papers on the assumption that they are timely, have a higher visibility quality and meet standards of novelty. In this study, about 57% of the work in disaster on ontologies development is presented as conference publications followed by 34% as journal papers while the rest are technical reports and any other documents.

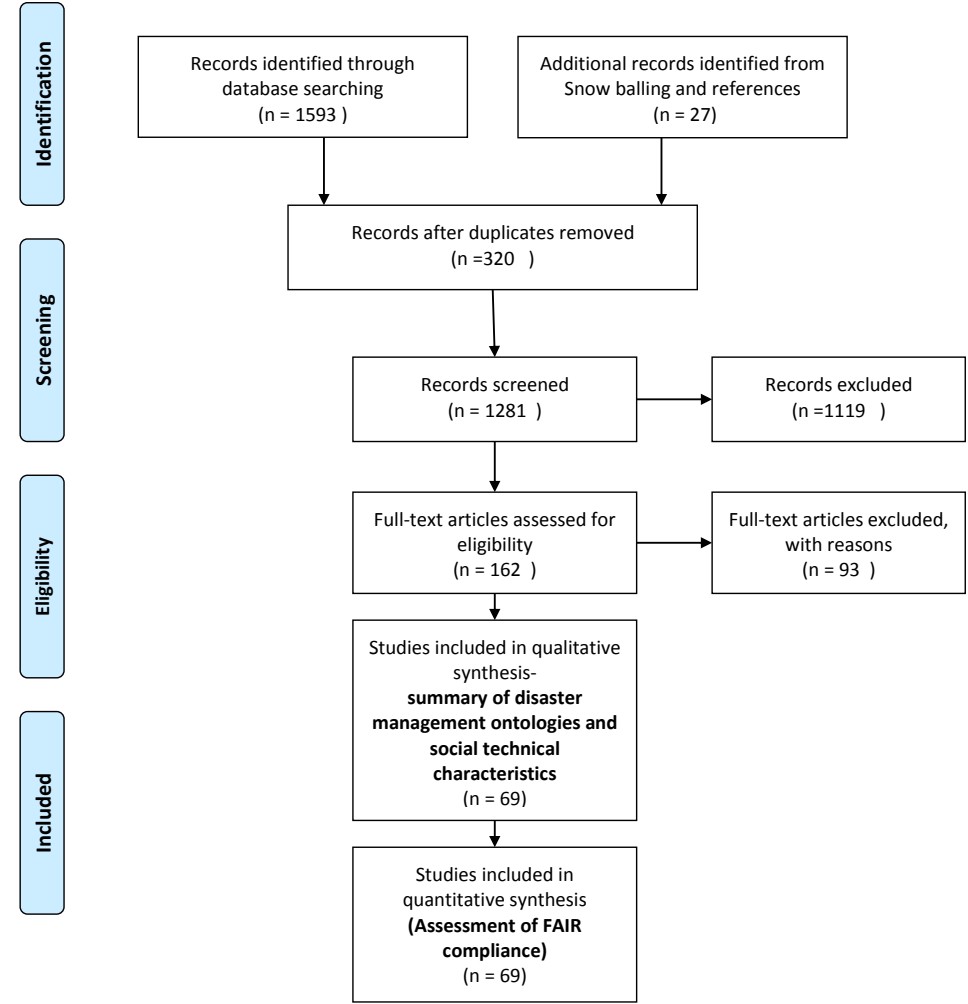

**Figure 1.** Search Results.

### 4.1. RQ1: Which Formal Ontologies Exist in the Disaster Domain?

Tables 4 and 5 show that a large corpus of literature exists on the development of formal ontologies for representing knowledge in the different sub domains in the disaster management cycle. A total of sixty nine (69) ontologies are extracted for all phases of the disaster management cycle(DMC). The response phase of the DMC constitutes the largest number (66.7%) of ontologies extracted, followed by the Hazard, vulnerability and Risk analysis phase with 23.2% while the rest (10.1%) represent the recovery, prevention, mitigation and early warning phases. The eminent ontology development activities in the response phase imply a complex and diverse knowledge base exists due to heterogeneous participants during humanitarian and crisis operations. A critical factor to developing ontology artefacts is experience and expertise towards shared understanding [35] within the development team. Results in Figure 2a indicate 63% of ontologies are developed by knowledge experts. Also 31% of the ontologies are developed by a heterogeneous team composed of both disaster management and knowledge experts. That is to say, 94% of teams have an expert with competences in knowledge representation using community standards and practices. This explains the high values for interoperability and reuse metrics in Section 4.2.

Only 6% of the development work is undertaken by disaster domain experts. Disaster management experts provide relevant conceptual knowledge but lack knowledge modelling skills. To overcome this limitation, Hammar [36] proposes the use of ODPs to enable domain experts create ontologies without support of knowledge engineers. Le Franc et al. [17], indicate that the reuse of ODPs is critical to improving FAIRness of ontologies.

**Table 4.** Disaster Response Ontologies (Crisis and Humanitarian Domain).

| ID | Ontology | Description |
|---|---|---|
| P57 | beAWARE ontology [37] | Ontology for integrating heterogeneous data in the context of climate crisis management. |
| P40 | Communication and Tracking Ontology [38] | For common understanding of communication and tracking operations by stakeholders in disaster relief |
| P28 | Disaster-domain-model [39] | Represents interdependencies between resources and needs during crisis management. |
| P23 | Crisis situation ontology [40] | Represents emergency shared situation awareness knowledge for stakeholders in crisis management |
| P69 | CROnto ontology [41] | Represents crisis features, impacts and strategic response plans between stakeholder organisations. |
| P10 | Disaster management domain ontology [2] | Provide a unified understanding of knowledge from heterogeneous glossaries in the disaster domain |
| P36 | Disaster management service ontology [42] | Models internal and external information for slope based landslide analysis. |
| P54 | Disaster medical relief ontology [43] | Structure common knowledge for disaster medical relief |
| P60 | Disaster Trail Management (DTM) [44] | Represents knowledge in the earthquake disaster response phase |
| P81 | DOcument-Report-Event Situation (Dores) [45] | Represents information collected about events, crises and representing event and situation reported |
| P63 | Disaster ontology [46] | Defines disaster risk knowledge in the context of mining social media data during disaster and crisis |
| P3 | Domain Ontology for Mass Gathering (DO4MG) [47] | Defines a knowledge model for representing emergency management concepts for mass gatherings |
| P25 | Earthquake ontology [48] | Defines concepts in the Earthquake domain for use in disaster management |
| P76 | EDXL_RESCUER [49] | Defines Emergency and Crisis domain knowledge necessary for coordination and exchange of information within legacy systems |
| P17 | Emergency case ontology [50] | Defines earthquake emergency response knowledge for decision support. |
| P16 | Emergency decision ontology [51] | Represents Natural disaster emergency knowledge to support decision making. |
| P77 | EMERgency Elements (EMERGEL) [52] | Defines emergency concepts for semantic mediation services in Emergency Management Systems. |
| P9 | Emergency Management Ontology [53] | Ontology for supporting semantic interoperability during emergency management |
| P79 | Emergency Response Ontology [54] | Represents fire and emergency response(FER) knowledge to support the FER indicators |
| P24 | Emergency Response Ontology [55] | Represents emergency situation knowledge in the context of Mobile-based emergency response systems. |
| P56 | EmergencyFire ontology [56] | Represents knowledge in fire emergency response situations |
| P85 | Empathi [4] | Provides concepts for integrating emergency management information from various sources such as satellite images, local sensors and social media content |
| P51 | EPISECC Ontology [57] | Spatio-temporal disaster management ontology that defines knowledge to first responders. |
| P66 | GEO-MD ontology [58] | A geographic ontology representing major disaster concepts used in satellite image classification |
| P47 | Geontology meteorological disaster ontology [59] | Formal definition of knowledge for emergency management of meteorological disasters |
| P43 | Humanitarian Aid for Refugees in Emergencies (HARE) [60] ontology | Enables integration of humanitarian aid information from several legacy databases |
| P29 | Humanitarian Assistance Ontology [61] | Represents Humanitarian Crisis knowledge for disaster response. |
| P74 | Humanitarian Exchange Language (HXL) [62] | Vocabulary developed by UNOCHA to improve data management and exchange for disaster response |
| P73 | Management Of A Crisis (MOAC) vocabulary [63] | Vocabulary for integrating crowd generated content with traditional information extracted from humanitarian assessment reports. |
| P46 | Meteorological disaster ontology (MDO) [64] | Describes components of the meteorological disaster to support emergency management. |
| P64 | Natural Disaster Ontology [65] | Represents disaster knowledge with for semantic extraction of disaster related online articles |
| P44 | Ont-EP4MO [66] | Defines knowledge for emergency management in metro operations |
| P49 | OntoCity [67] | Incoporates refactoring of the spatial relations for reasoning upon imagery data for disaster management |
| P38 | OntoEmerge ontology [68] | Defines emergency planning knowledge |
| P11 | OntoFire ontology [69] | Wildfires ontology for enriching of data to enable search and retrieval from the ontofire Geoportal. |
| P83 | Ontology for climate crisis management [70], | Represents knowledge about mission assignments to first responder units during a climate crisis event |
| P61 | Ontology for flood fore casting [71] | Captures flood response knowledge for shared understanding among flood response stakeholders |
| P84 | POLARISC Ontology [72] | Represents shared knowledge amongst emergency responders in the disaster response process. |
| P50 | PS/EM Communication [73] | Capture semantics in enterprise public safety and emergency management systems |
| P65 | ResOnt [74] | Supports Common data interpretation for french firefighters participating in rescue operations |
| P5 | Simple Emergency Alerts for All (SEMA4A) [75] | Represents knowledge for emergency notification systems accessibility |
| P41 | Social Media Emergency Management (SMEM) [76] | Ontology that links social media data with emergency related information. |
| P33 | SOFERS [77] | Ontology for managing scenario information and disaster conditions during Emergency Response |
| P14 | SOKNOS [78] | Core domain ontology for representing emergency management knowledge |
| P15 | Typhoon Disasters Ontology [79] | Defines concepts for representing knowledge about typhoon disaster |
| P2 | Web Based ontology structure (WB-OS) [80] | Structure knowledge in web-based natural disaster management systems |

**Table 5.** HVR, Preparedness, Recovery and other DRR Ontologies.

| ID | Ontology | Description |
|---|---|---|
| | **HAZARD, VULNERABILITY AND RISK (HVR) ANALYSIS PHASE** | |
| P58 | Chemical ontology [81] | Pattern-based ontology defining chemical processes and associated hazard classifications |
| P8 | Flood risk assessment ontology [82] | Represents flood risk assessment knowledge and stakeholder requirements |
| P37 | Flood risk assessment ontology [83] | Represents knowledge and processes for flood risk assessment derived from different perceptual models of watershed flood risks. |
| P53 | Flood Scene Ontology (FSO) [84] | Defines spatial-contextual semantics of the flood disaster for disaster management using the context of mining knowledge from satellite imagery |
| P26 | Flood Ontology [85] | Represents flood forecasting knowledge based on continuous measurements of water parameters |
| P4 | Geological Hazard ontology [86] | Defines concepts for representing knowledge about geological hazards |
| P39 | Hazard causation ontology [87] | Defines hazards, their causes, consequences and relationships between them. |
| P78 | Hazardous situation ODP [30] | Ontology design pattern(ODP) for modelling hazardous situations |
| P82 | InfraRisk ontology [88] | Represents knowledge about natural hazard events and their impact on the infrastructure component |
| P80 | Modified Hazardous Situation ODP [29] | Modifies the existing HazardousSituation ODP to support risk assessment and mitigation planning concepts. |
| P21 | MONITOR ontologies [89] | Defines modular ontologies for disaster risk management developed under the MONITOR EU project. |
| P71 | NNEW weather ontology [90] | Defines knowledge for representing weather observation |
| P75 | Ontology for Vulnerability Assessments [91] | Ontology for Vulnerability Assessments implemented in the VUWIKI |
| P62 | Ontology model for hazard identification [92] | Models knowledge used in rapid risk estimation for hazard scenarios |
| P86 | QualityCausation [93] | Represents causation of qualities of an object that participates in a hazard event |
| P72 | Referential quality ODP [94], | Represents knowledge for qualities(such as notions affordance, resilience and vulnerability) of an entity with reference to an external factor |
| | **RECOVERY PHASE** | |
| P30 | Disaster domain ontology [95] | Concepts based on Critical GEOS Earth Observation Parameters and Social Benefit Area |
| P48 | Dynamic Flood Ontology (DFO) [96] | Models spatial-temporal changes of flood situation for disaster monitoring purposes |
| | **PREVENTION, MITIGATION AND PREPAREDNESS PHASES** | |
| P70 | Disaster resilient construction operations (DRCOs-Onto) [97] | Defines knowledge for whole life-cycle disaster management of construction projects |
| P67 | Landslip Ontology [98] | Unified knowledge representation for EO data discovery of during landslide hazard verification and analysis in EWS. |
| P59 | SWRO-DDPM ontology [99] | Defines concepts for sensors, observation and model resources for dynamic disaster processing |
| P45 | Urban Industrial Disaster Warning ontology [100] | Represents knowledge for technology event in the context of urban industrial disaster warning. |
| | **OTHERS** | |
| P1 | Ontology for DRR learning resources [101,102] | Enhance the sharing of knowledge and learning about disaster risk reduction. |

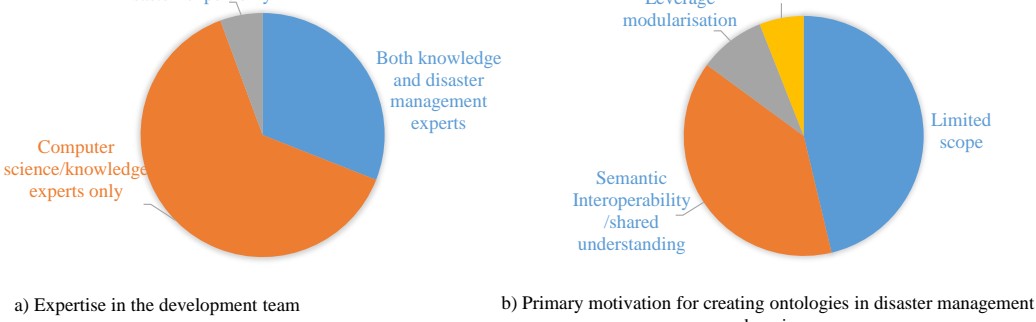

a) Expertise in the development team

b) Primary motivation for creating ontologies in disaster management domain

**Figure 2.** Summary of Expertise of Development Team and Motivation for Ontology Development.

There exist minimal efforts for developing disaster management knowledge using the bottom-up approach (7.2%) to ontology development. In the entire disaster management cycle, bottom-up development exists only in the hazard, vulnerability and risk (HVR) phase (see Figure 3). Such reusable ontologies for HVR modelling include the MONITOR ontologies [89] and ODPs such as hazardous situation [29,30], chemical process [81], referent qualities [94]. The remaining 92.8% of the ontologies are developed using the top down approach thus contributing to the low FAIR value.

In Figure 2b ontologies in the disaster management domain are developed with the motivation of solving scope, expressivity, interoperability and modularisation problems. The development of new ontologies in the disaster management domain is mostly motivated by the fact that existing ontologies do not fit the scope of tasks at hand. The scope

issue is most prominent in the emergency/humanitarian response and relief since it is a complex activity involving many actors with different knowledge requirements [53].

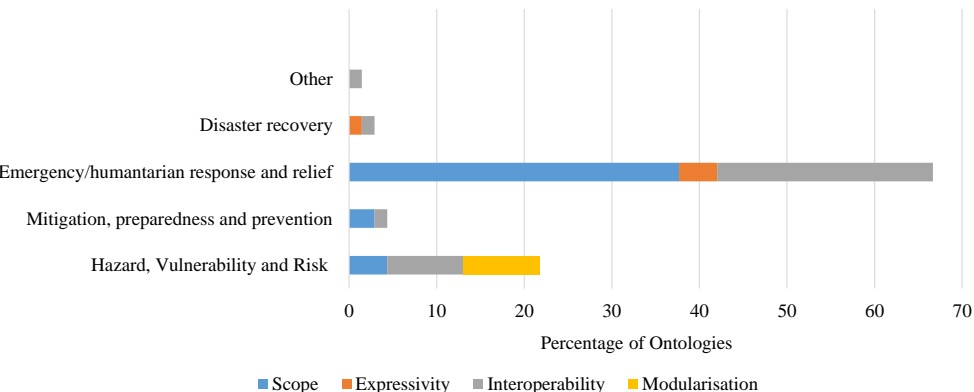

**Figure 3.** Motivation for Developing Ontologies within the Disaster Sub-domains.

Also, in Figure 3, results reveal that limited expressivity is a motivating factor for creating ontologies in the emergency response and disaster recovery subdomains. Within the FAIR Framework, PRec. 11 recommendation proposes standards for handling expressivity to ensure interoperability. However, ODPs (BPRec3) as a best practice can be applied to solve the scope, expressivity problem and foster interoperability in the disaster management domain.

### 4.2. RQ2: To What Extent Are Formal Ontologies in the Disaster Management Domain FAIR?

This section presents the extent to which formal ontologies for representing disaster knowledge are Findable, Accessible, Interoperable and Reusable. Figure 4 presents the resulting FAIR metric values for ontologies in the disaster domain drawn from publications.

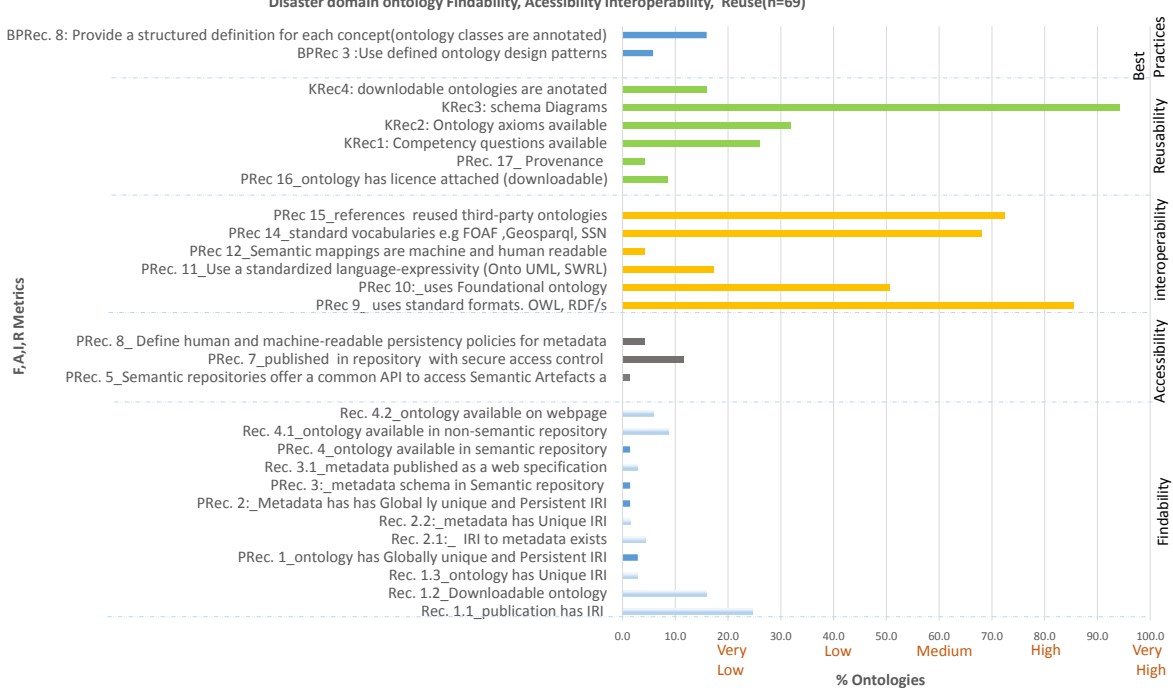

**Figure 4.** Adherence to FAIR principles for Ontology Artefacts within the Disaster Management Domain .

### 4.2.1. Findability

While Tables 4 and 5 confirm existence of a substantial number of ontologies for representing disaster management knowledge, results for PRec.1 show that only 24.6%(17) have URIs (see Figure 4) to disaster management ontology artefacts.

However, only 15.9%(11) of all the disaster management ontologies in Tables 4 and 5 have resolvable URIs. This means that there exists ontologies whose physical artefacts are not findable due to the fact that no URI to the resource is available while the rest are not resolvable possibly due to non-persistence. Further investigation of downloadable ontology artefacts in Table 6, reveals that 90.9% of the downloadable disaster ontology artefacts have identifiers that do not conform to the principle of uniqueness and persistence. This is because the URIs either directly point to files or local name-spaces pointing to the content of the files hosted on web pages or Github.

**Table 6.** Ontologies with URIs in the Disaster Domain.

| ID | Ontology | URI To File, accessed on 21 December 2020 | URI Resolves | Concepts | Properties | Unique URI | Persistent URI | Metadata Specification Available | Published in Repository |
|---|---|---|---|---|---|---|---|---|---|
| 1 | Disaster-domain-model | https://tinyurl.com/DisasterModel | No | - | - | No | No | No | No |
| 2 | Dores | https://tinyurl.com/DoresOnt | No | - | - | Yes | No | No | No |
| 3 | EDXL_RESCUER | http://www.rescuer-project.org/ | No | - | - | No | No | No | No |
| 4 | EMERGEL | http://vocab.ctic.es/emergel/ | No | - | - | No | No | No | No |
| 5 | Emergency response ontology | http://ontology.eil.utoronto.ca/GCI/FireEmergency/EResponse.owl | Yes | 32 | 17 | No | No | No | No (weblink) |
| 6 | empathi | https://tinyurl.com/EmpathiOwl | Yes | 423 | 338 | Yes | Yes | Yes | No (webpage) |
| 7 | Hazardous situation ODP | https://tinyurl.com/HazardousS | Yes | 9 | 8 | No | No | No | yes (ODP portal github) |
| 8 | Humanitarian Exchange Language (HXL) | https://tinyurl.com/HXLttl | Yes | 50 | 66 | No | No | Yes | yes (github) |
| 9 | InfraRisk ontology | https://tinyurl.com/infrariskttl | Yes | 33 | 48 | No | No | Yes | No (Webpage) |
| 10 | Management Of A Crisis (MOAC) vocabulary | https://tinyurl.com/MoacVocab | Yes | 92 | 32 | No | yes | yes | Yes (LOV, BARTOC) |
| 11 | Modified Harzardous situation ODP | https://tinyurl.com/MHazardousODP | Yes | 13 | 18 | No | No | No | Yes (github, ODP portal) |
| 12 | NNEW weather ontology | https://wiki.ucar.edu/display/CSSWX | No | - | - | No | No | No | No |
| 13 | Ontology for Vulnerability Assessment | www.vuwiki.org | No | - | - | No | No | No | No |
| 14 | Ontology for climate crisis management-beWARE | https://goo.gl/5VR4qB | yes | 63 | 83 | No | No | No | Yes (github) |
| 15 | POLARISC Ontology | https://tinyurl.com/POLARISCO | Yes | 9807 | 281 | No | No | No | Yes (github) |
| 16 | Qualitycausation | http://w3id.org/gicentre/onto/QualityDependenceCompositionData | Yes | 93 | 123 | yes | yes | No | Yes (github) |
| 17 | Referential quality ODP | https://tinyurl.com/referentialOdp | Yes | - | - | No | No | No | No |

Metric PRec.2 requires that the metadata record for the ontology artefact should use globally unique and resolvable identifiers. Only 4.3%(3) of all the retrieved ontologies in Table 6, have links to their metadata specifications as shown in Table 7. However, only the empathi ontology metadata specifications has a URI that is unique and persistent. The remaining ontology metadata records have URIs either directly pointing to files or local namespaces pointing to the content of the files hosted on web pages.

**Table 7.** Metadata Specifications for Ontology Artefacts within the Disaster Domain.

| | Ontology | URI To Metadata, accessed on 21 December 2020 | Metadata Specification Resolvable | Unique URI | Persistent URI | Published in Repository |
|---|---|---|---|---|---|---|
| 1 | Management Of A Crisis (MOAC) vocabulary | http://observedchange.com/moac/ns/ | Yes | No | No | BARTOC LOV |
| 2 | Empathi | https://w3id.org/empathi/ | Yes | Yes | Yes | No (webpage) |
| 3 | InfraRisk ontology | http://vocabs.datagraft.net/infrarisk | Yes | No | No | web specification |

A further investigation of the metric PRec.3 indicates that metadata specifications in Table 7 are published in line with OBO foundry specifications.

Recommendation PRec.4 requires publication of disaster domain ontology artefacts/content in a semantic repository. Results in Table 6, reveal that only 1.4%(1) of all the retrieved ontologies are published in semantic repositories (i.e., Bartoc and LOV). The remaining resolvable ontologies that total to 11.6%(8) of total retrieved ontologies are published in non-semantic repositories (such as Github) or websites. This implies that while there exists vast ontologies for representing disaster knowledge(see Tables 4 and 5), about 84.1% of the artefacts are not published at all, while a handful are published in non semantic repositories such as github that present findability barrier due to the inability to parse and download the ontology content (i.e., concepts/terms, relations and metadata). Generally, results in Figure 4, and Equations (4) and (5) reveal that ontology artefacts representing disaster knowledge have a very low findability i.e., on average 1.775% of total ontology artefacts are findable.

$$Findability\_value = \frac{W_i * (PRec:1 + PRec:2 + PRec:3 + PRec:4)}{4} \tag{4}$$

$$Findability\_value = \frac{1 * (2.9 + 1.4 + 1.4 + 1.4)}{4} = 1.775(very\_low) \tag{5}$$

To better understand the extent of ontology findability in line with PRec: 4, a search of all repositories in the Bartoc registry and Github yielded a small number of disaster management ontologies (5) as listed in Table 8.

**Table 8.** Ontologies Extracted from Repository Search.

| Ontology | URI, accessed on 21 December 2020 | Source (Repository) |
|---|---|---|
| Management of a Crisis Vocabulary | http://www.observedchange.com/moac/ns | LOV, BARTOC https://bartoc.org/, accessed on 21st December 2020 |
| Vocabulary to describe incident response by emergency services | https://lov.linkeddata.es/dataset/lov/vocabs/incident/versions/2015-06-22.n3 | http://vocab.resc.info/incident, accessed on 21st December 2020 BARTOC |
| RiskHackathon-risk ontology | https://github.com/MikeHypercube/RiskHackathon | Github |
| rioter-risk-ontology | https://github.com/rioter-project/rioter-risk-ontology | Github |
| EAonto | https://github.com/julian-garrido/EAonto | Github |
| Disaster ontology | https://onki.fi/en/browser/overview/disaster | ONKI portal /Finto service |

The study reveals that ontology artefacts are published in generic repositories and could not find any repository that is dedicated to archiving disaster domain knowledge. Through collaborative efforts, community stakeholders (i.e., development agencies, Governments and researchers, etc.) need to implement disaster domain-specific semantic repositories such as that in [103] where disaster knowledge is published, documented and indexed. This presents the possibility of managing the evolving nature of disaster knowledge as well contribute to its findability.

### 4.2.2. Accessibility

The recommendation PRec: 5 requires that ontologies are published in semantic repositories that offer a common access API (see https://bartoc.org/registries, accessed on 11 January 2021) to access ontology artefacts and there content in several formats. Results reveal that only 1.4% (1) of retrieved ontologies in the disaster domain are published in semantic repositories as shown in Table 6 that are in line with PRec: 5 (i.e., MOAC is published in LOV and Bartoc).

As already noted in Figure 4, 11.69% of the total retrieved ontologies are published in a repository(both semantic and non semantic). All these repositories have secure protocol and user access control functionalities as recommended in PRec: 7

$$Accessibility\_value = \frac{W_i(PRec:5 + PRec:7 + PRec:8)}{3} \tag{6}$$

$$Accessibility\_value = \frac{1 * (1.4 + 11.6 + 4.3)}{3} = 5.77 \tag{7}$$

Based on retrieved ontologies from literature, we can conclude that accessibility for ontologies in the disaster domain is very low. The flexibility with which ontology artefacts are accessible has an impact on other Findability, interoperability and reuse principles of the FAIR guidelines as discussed in Section 5.5.

### 4.2.3. Interoperability

Recommendation PRec.9 requires that ontology artefacts are represented using common serialization formats. Results in Figure 4 reveal a very high number (85.9%) of the disaster management ontology artefacts are serialized using semantic web and linked data standards such as RDF, RDFs, various OWL profiles, etc. The remaining publications only present ontologies as schema/tree diagrams.

To enable interoperability between disaster ontologies, the PRec.10 requires that artefacts use a Foundational Ontology to align semantic artefacts. In Figure 4, results reveal that 50.7% (35) of the retrieved disaster domain artefacts in literature are explicitly developed u sing an existing foundational ontology to ground disaster domain concepts based on specific world of discourse such as DOLCE, BFO, UFO, etc as visualized in Figure 5.

To express characteristics of more complex/expressive semantic models, results reveal that 17.4% of artefacts represent disaster ontology language using additional common languages such as SWRL, OntoUML in line with PRec.11 recommendation. To achieve recommendation PRec.12, Semantic mappings between the different elements of semantic artefacts should use machine-readable formats based on W3C standards. Results reveal 4.3% (3) of the retrieved disaster management ontologies detail explicit mapping/correspondences between ontologies.

In addition, 68.1% of the disaster domain ontology artefacts are inline with PRec.14 recommendation that encourage the use at least one of the standard vocabularies (like FOAF, Geosparql, schema.org, Dublin core, Prov-O) to describe disaster domain ontologies. 72.5% of the ontologies the disaster reuse 3rd party ontologies thus contributing to interoperability.

$$Interoperability\_value = \frac{W_i(PRec:9 + PRec:10 + PRec:11 + PRec12 + PRec14 + PRec15)}{6} \qquad (8)$$

$$Interoperability\_value = \frac{1 * (85.9 + 50.7 + 17.4 + 4.3 + 68.1 + 72.5)}{6} = 49.1\% \qquad (9)$$

From Equation (9), we can conclude that disaster domain ontologies have moderate adherence to interoperability recommendations since the help of knowledge experts is highly utilized in the development of these artefacts as indicated in Figure 2. These experts typically having a good working knowledge of community standards and recommendations. Beyond FAIR metrics, future work can be done to provide an understanding of interoperability of the resolvable artefacts through concept matching and reasoning about correspondences within the disaster management domain ontologies.

### 4.2.4. Reusability

Results in Appendix A Table A1 reveal that several existing vocabularies are reused in developing formal ontologies in the disaster domain. To gain a bigger picture of reuse in the disaster domain, the word cloud in Figure 5 shows vocabularies reused to develop formal ontologies. Reused Vocabularies range from glossaries, thesauri to formal ontologies. Glossaries and thesauri are typically used to draw out disaster domain knowledge. From Figure 5, the EM-DAT is most used of all other glossaries. Furthermore, foundational ontologies such as SWEET, DOLCE, BFO, SUMO UFO as well as a number of standard ontologies (such as SKOS, FOAF, DC-Terms, OWL-Time, GeoSPARQL, SSN) are most reused in the development of disaster management ontologies.

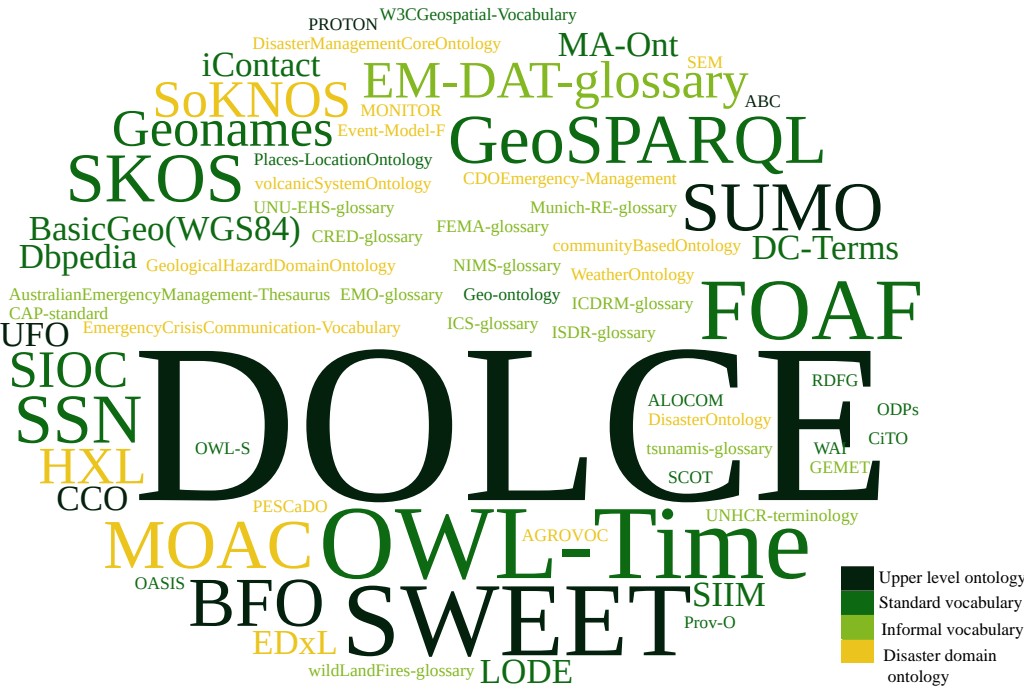

**Figure 5.** Tag Cloud of Reused Vocabularies for Creating Ontologies within the Disaster Domain.

At a disaster management domain level, vocabularies such as MOAC, HXL, SoKNOS are most reused to develop new ontologies. This could partly be attributed to the fact that these ontologies and/or their documentations are available in repositories to support reuse. Further, investigation of downloadable disaster domain ontologies shown in Table 9 (see *Data Availability Statement*) reveals that all resolvable ontologies retrieved from literature directly import external ontologies using the *owl:imports* construct. Results show that 8.7% (6) of the reviewed ontology artefacts in the disaster domain have open-source licenses as recommended by PRec.16. These licenses include Creative Commons 4.0, GNU Lesser General Public License and MIT license. Licenses enable legal interoperability by making reuse rights explicit for machines and humans.

Results also reveal that about 4.3% of all the disaster domain ontologies use standard models such as PROV-O to document provenance information. This information is critical in evaluating the disaster ontology artefact and understand its release cycle. The low number of PRec: 16 and 17 is attributed to the fact that it is only possible to check descriptions in physical/downloadable artefacts unless stated in the publication. Based on indicators of re usability in [25], literature revealed that 94.6% of the retrieved disaster ontologies have schema diagrams, 31.9% have axioms available, while 26.1% have competency questions.

$$Reusability\_value = \frac{W_i(PRec:16 + PRec:17 + R1 + R2 + R3 + R4)}{6} \quad (10)$$

$$Reusability\_value = \frac{(8.7 + 4.3 + 26.1 + 31.9 + 94.2 + 15.9)}{6} = 30.2\% \quad (11)$$

Not only do most ontologies in disaster management hardly reuse concepts from standard ontologies, their concepts are also rarely reused by disaster domain ontologies as shown in Figure 5. This generally explains a fairly low reusability value in Equation (11).

**Table 9.** Comparison of Resolvable Ontologies within the Disaster Domain.

| Ontology | Format | Import | Annotations Property | Provenance | License | IRI_VERSION |
|---|---|---|---|---|---|---|
| Management of a Crisis Vocabulary MOAC | rdf(n3) | none | dcterms, foaf, owl, rdf | Yes-S | GNU Lesser General Public License. Specification | Yes (S*) No (O) |
| Empathi | owl | disasterModel.owl Geonames ontology icontactOwl dcterms foaf, olia.owl sioc, ma-ont | dcterms, rdfs | Yes-S*, O* | (CC By 4.0) Specification | yes (L, S*) |
| Humanitarian Exchange Language (HXL) | ttl | none | Foaf, dcterms, rdfs, skos | none | CC BY 3.0-(O) | |
| POLARISC Ontology | owl | AllCoreOntology, PollariscHealthCareResources PCC PolariscO PolariscFighters PolariscGendarmerie PolariscPublicAuthorities PolariscHealthCareUnits PolariscPolice Polarisc Messages –Indirect imports BFO, CCO InformationEntityOntology ExtendedRelationOntology AgentOntology ArtifactOntology eventOntology QualityOntology Geospatial ontology timeOntology UnitOfMeasure OBOfoundry ro diod non-classified | rdfs, foaf,dcterms | none | none | None |
| EResponse Ontology | owl | none | dcterms, rdfs, | none | CC BY 3.0 (O) | none |
| ontology for climate crisis management (beWARE) | owl | skos core | dcterms, rdf, rdfs, skos owl | none | none | Annotation |
| InfraRisk ontology | ttl/rdf | none | dc terms, foaf, rdfs | none | CC BY 3.0 (O*, S*) | none |
| Hazardous situation ODP | owl | Cp annotation schema | CP annotation schema owl | none | none | Annotation 1.0 |
| Modified Hazardous Situation ODP | owl | 1-Time interval 2-CP Annotation schema | CP annotation schema rdfs annotation owl | None | None | Annotation 1.0 |
| Referent Quality ODP | owl | ExtendedDns | dcterms, rdfs, skos | | | |
| Quality Dependence (Vulnerability) | owl | DUL, CP annotation schema | cp annotation schema, owl | none | none | Annotation 1.0 |
| Risk ontology (Risk Hackathon) | owl | Event, Time, skos-core, OMG specificationMetadata OMG annotationVocubulary, OMG Goals | dcterms, rdfs, owl | None | MIT licence | None |
| Disaster ontology | rdf | none | dcterm, rdfs | None | None | None |
| Incident ontology | n3 | none | yes (dc terms) | None | CC0 1.0 (O*, S*) | None |
| Risk ontology (Rioter) | owl | none | dcterms | None | MIT License | Annotation 1.0 |
| EIA ontology | owl | none | rdfs | None | None | None |

O* extracted from ontology, S* extracted from specification.

### 4.2.5. FAIR Best Practice Recommendations

To assess adherence of ontologies to FAIR best practice recommendations in the disaster management domain, we use BPRec:2 and 8. Results reveal that only 5.8%(4) of the retrievable ontologies use ontology design patterns in the development of disaster domain ontologies.

$$FAIR\_BestPractices\_value = \frac{W_i(BPRec3 + BPRec8)}{2} \tag{12}$$

$$FAIR\_BestPractices\_value = \frac{(5.8 + 15.9)}{2} = 10.85\% \tag{13}$$

We randomly sample seven (7) ontologies from Tables 4 and 5 and explore whether ODPs can be used to represent disaster domain knowledge(Using CQs, schema diagrams and physical artefacts). In Table 10, shows ontologies and respective ontology design patterns that can be reused to better characterise the ontology. This provides evidence that ontology developers in the disaster management domain generally do not reuse existing ODPs. For instance ontologies 1-6 show the possibility to reuse patterns in the humanitarian and crisis domain. Reuse of ODPs is critical to improving quality and interoperability [104] of disaster management ontologies.

**Table 10.** Examples of Ontologies that can be Re-characterised by ODPs.

| S/N | Ontology | Proposed Patterns |
|---|---|---|
| 1 | Emergency case ontology [50] | Organisation, AgentRole, Location, event, informationObject |
| 2 | crisis situation ontology [40] | Event, Organisation |
| 3 | Disaster Trail Management (DTM) [44] | Event agent role and organisation |
| 4 | ResOnt [74] | ObjectRole, Task, organisation |
| 5 | Empathi [4] | Event, place, object participant |
| 6 | geontology meteorological disaster [59] | Event, Event causation, Place can be aligned to Geosparql |
| 7 | Infrarisk [88] | event, place |

In Figure 3, the development of new artefacts is primarily motivated by the desire to solve the problem of limited scope, semantic interoperability barrier(shared understanding), limited expressivity as well as leverage benefits of modularisation. ODPs provide the ability to reuse existing disaster knowledge while solving these problems. Future work could focus on extracting content ODPs from legacy ontologies in the disaster domain.

BPRec:8 proposes a structured definition for each concept as a best practice for FAIR ontologies. A further analysis also reveals that 15.9% (11 i.e., 100% resolvable), of the retrievable disaster management ontologies are annotated at concept level making it easy for users to understand them.

Table 11 presents a summary of FAIR principle values. Generally, the average value for FAIR components indicates that a very low FAIRness for disaster management ontologies. Given the recursive definition of interoperability in the FAIR framework [10], a very low FAIR ontology Value contributes to low data interoperability maturity for linked data in the disaster management domain. The FAIR value in Table 11 is dependent on metrics that are entailed in Equation (1). Since we cannot evaluate some metrics, the FAIR calculated is a view of FAIRness based on a set of defined metrics.

**Table 11.** Summary of FAIR Principles for Disaster Management Ontologies.

| S/N | Principle | No. of Indicators | Value (%) | Classification |
|---|---|---|---|---|
| 1 | Findability | 4 | 1.8 | Very low |
| 2 | Accessibility | 3 | 5.8 | Very low |
| 3 | Interoperability | 6 | 49.1 | Moderate |
| 4 | Reusability | 6 | 30.2 | Low |
| 5 | Use of Best Practices | 2 | 10.8 | Very low |
|  | FAIR (Average) |  | 19.54 | Very low |

## 5. Discussion of Results

Results presented in Section 4 show a low Findability and accessibility for disaster management knowledge. Also, disaster knowledge has moderate interoperability and reusability. Ontology findability, access [105] and reuse in the biomedical domain [106,107] are slightly higher than in the disaster domain because of ontologies are archived using established design principles (e.g., the OBO foundry principles) in dedicated semantic repositories. The ability to share and discover knowledge using a coordinated approach towards semantics will guarantee the reuse of evolving disaster management knowledge and data interoperability. Furthermore, this study provides insights into tradeoffs in FAIR Principles for optimization of disaster data interoperability API. Below we present a detailed discussion of implications for research and industry.

### 5.1. Need for Disaster Community Repository That Support Evolution of Disaster Knowledge

Sustainable comparison of disaster knowledge requires older versions of ontologies that available via a persistent URI [14]. Tables 4 and 5 contain a number both task and domain ontologies for disaster management. These present evidence of the wide use of ontologies for organising data and knowledge in the disaster domain. However, in Section 4.2.1, a large percentage of these ontologies are not findable given that they have no GUPRI and neither published in existing semantic repositories. Disaster-related development agencies maintain glossaries and thesauri for disaster management [108,109]. To support the Findability of disaster ontologies in line with FAIR principles, these organisations can extend support to the development and maintenance of curated repositories focused on archiving disaster ontology artefacts. This initiative similar to the biomedical domain [103] and Libraries of curated ontologies in [110] ensures coverage and reuse of evolving disaster knowledge. These repositories could contain already proposed features by existing FAIR metrics for semantic artefacts (e.g., community based features for ontology annotation and evaluation enabling disaster domain users to provide feedback [17] as well as newly identified metrics. The ability for disaster domain experts to find and reuse ontologies and best practices for organising disaster data will in turn lead to interoperability.

### 5.2. Identifying Metrics for FAIR Principles

As already suggested, the definition of best practice for the semantic web domain is not a new concept, but are rather holistically redefined by the FAIR Principles. However, there is not a known complete list of FAIR maturity indicators/ best practices that can be used to assess FAIRness of Disaster Management Ontology. While this work uses metrics and best practices from [17,25], future work could focus on identifying more FAIR indicators for semantic artefacts. These metrics are key in FAIR maturity assessment for disaster management knowledge as well as guide domain and industry experts in reviewing disaster management ontology before it is published in academic and industry repositories.

### 5.3. Automated Evaluation of Ontology FAIR Metrics

In this study, we evaluate the FAIRness of a disaster ontology artefact using a set of semi-quantitative metrics. However, not all metrics can be assessed due to non-applicability to the artefact. Other indicators are related which makes objective evaluation difficult. This coupled with the fact that there is an unknown complete list of FAIR metrics available implies that the FAIR assessment score does not provide an intrinsic value but rather a picture for FAIRness of disaster management ontology based on metrics used. Therefore, future work could be undertaken to develop an objective, intuitive and easy to use apps that automate FAIR assessment tools. Although Work in this direction has been proposed by Wilkinson et al. [111], customised tools for assessing FAIR maturity by the semantic community could be undertaken.

### 5.4. Compilation of Best Practices and Use Cases for Disaster Community

Disaster Management Ontologies are predominantly developed by knowledge experts, a combination of knowledge and domain experts (see Figure 2). This partly explains why some interoperability and reuse metrics are high since knowledge experts have an understanding of community standards/practices. Results also reveal that modular approaches such as ODPs are reused minimally in the disaster domain. Therefore, ontology development in the disaster management domain often requires the assistance of a knowledge engineer. In Table 10, we show that existing ontologies developed based on a top-down approach can be represented by ODPs. ODP *reuse* is a powerful tool for ensuring domain experts can organise data without the help of knowledge experts. Also, ODPs are key in improving *interoperability* while solving expressivity and scope issues presented in Figure 3 in the disaster management domain. Therefore, ODPs serve as best practices for improving interoperability and reuse in the FAIR framework. Adoption of ODPs requires the existence of patterns that can answer relevant disaster domain requirements. Besides, a catalogue of highly visible use cases for disaster management is a powerful driver of FAIRification efforts by the disaster community.

### 5.5. Trade-Offs and Relationships in FAIR Components

The FAIR maturity evaluation reveals that disaster management ontologies are far from being Findable, Accessible and reusable. Ontology design patterns are proposed as best practices for improving FAIRness of Disaster management domain Ontology. Therefore, there exists a direct relationship between the use of ODPs and improved ontology interoperability and reuse. Indirectly, Ontology design pattern views proposed by [112] affect schema accessibility which in turn affects data accessibility as illustrated in Figure 6. Also, the concept of Interoperability in the FAIR framework is defined recursively, in such a way that data interoperability is achieved through the use of FAIR vocabularies/ontologies.

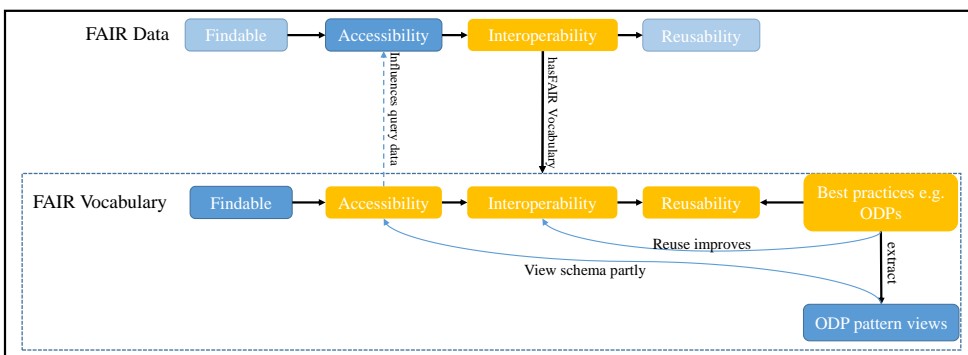

**Figure 6.** Example of Relationships in FAIR Principles.

Understanding relations in FAIR principles is critical for design prioritization for interoperability APIs in FAIR data stations. For instance, an API that prioritizes accessibility could incorporate pattern views which negatively affect how much of the schema/data are accessed for reuse and inter-operation. On the other hand, such an API provides sustainable access to ontologies that will increase the disaster management practitioners' confidence to reuse them. Therefore, future work could explore more relationships and trade-offs in the FAIR principles.

## 6. Limitations of the Study

This SLR has not been able to identify literature outside the search period. As such, we disregard disaster management ontologies developed before or after the search period in the analysis. Examples of such ontologies include SIADEX [113], ISyCri [114], BACAREX [115], ERO-M [116], etc. The other limitation of this work is that it presents results for formal

disaster management ontologies only. We do not consider informal vocabularies such as taxonomies, glossaries (e.g., EM-DAT, UNDRR, Reliefweb), etc.

Nevertheless, the SLR study allows us to present an insight into (1) the implementation of FAIR guidelines for managing disaster management ontologies and (2) implications for research and industry. Future work could focus on understanding and improving the Findability, Accessibility, Interoperability and reuse of informal disaster management vocabulary.

## 7. Conclusions and Recommendation

The difficulty in finding, accessing and reusing interoperable vocabularies to organise disaster management data creates a challenge for collaboration among stakeholders in the disaster management cycle on data integration tasks. This study set to assess the extent to which formal ontologies developed in the disaster management field are FAIR. In this paper, we extract 69 formal ontologies in the disaster management cycle. The highest number of these ontologies exists in the humanitarian and crisis response sub domain, followed by the HVR and preparedness sub domains. Besides, most disaster management ontologies are developed using a top-down approach. The low usage of ODPs implies that disaster domain experts will always find it hard to develop ontology artefacts without the help of knowledge engineers. Moreover, ODP reuse is critical for increasing interoperability and reuse for data. Results also indicate a very low findability, accessibility and reuse for disaster existing ontologies. However, some ontology interoperability practices are moderately used since most ontology development teams have knowledge experts on them. To conclude, ontologies in the disaster domain are far from being FAIR. To improve the FAIRness of disaster domain ontologies, stakeholders need to develop domain-specific repositories, identify metrics and tools for an automated FAIR assessment. Also, the community needs to compile best practices and FAIR use-cases which are powerful drivers of FAIRification efforts. Finally, our research reveals the importance of understanding tradeoffs in FAIR principles for API development in FAIR Data stations.

**Author Contributions:** Conceptualization, Imed Hammouda and Allan Mazimwe; methodology, Imed Hammouda and Anthony Gidudu; software, Allan Mazimwe; validation, Imed Hammouda, Anthony Gidudu; data curation, Allan Mazimwe; writing—original draft preparation, Allan Mazimwe; writing—review and editing, Imed Hammouda and Anthony Gidudu; supervision, Imed Hammouda and Anthony Gidudu. All authors contributed equally to the work. All authors have read and agreed to the published version of the manuscript.

**Funding:** This work is supported by the Swedish International Development Agency (SIDA) funding to Makerere University, Kampala in partnership with Chalmers University of Technology and Gothenburg University, Sweden under the BRIGHT project 317. The APC is funded by the BRIGHT 317 project.

**Institutional Review Board Statement:** Not applicable.

**Informed Consent Statement:** Not applicable.

**Data Availability Statement:** In this study, we provide data sets used in the SLR. First, we present a list of selected articles used in the review. The list contains article titles uniquely identified by ID of the series P** authors and year of publication. The data set also lists ontologies developed and other ontology characteristics (such as Format, the existence of URL, CQs, schema diagram, axioms, reused ontologies, reviewed ontologies, motivation for ontology creation). A list of Selected articles used in the review is downloadable from https://tinyurl.com/FairDisasterVocabulary, Secondly, we archive all downloadable disaster management ontology artefacts in a GitHub repository. This data is downloadable from https://tinyurl.com/FindableDisasterOntologies, We also archive the data set used for generating Tag Cloud Visualization in Figure 5 on Github. This data set is downloadable from https://tinyurl.com/FAIRtagcloud (all accessed on 2 January 2021).

**Conflicts of Interest:** No conflict of Interest exists. The funders had no role in the design of the study; in the collection, analysis, or interpretation of data; in the writing of the manuscript; or in the decision to publish the results.

## Abbreviations

The following abbreviations are used in this manuscript:

| | |
|---|---|
| ACM | Association for Computing Machinery |
| ALOCOM | Abstract Learning Object Content Model |
| APIs | Application Programming Interfaces |
| BACAREX | heavyweight ontology of planning objects and activities |
| BFO | Basic Formal Ontology |
| CAP | Common Alerting Protocol |
| CCO | Common Core Ontology |
| CDO | Core Domain Ontology |
| CiTO | Citation Typing Ontology |
| CP | Content ontology design pattern |
| CRED | Centre for Research on the Epidemiology of Disasters |
| CQs | Competency Questions |
| DCAT | Data Catalog Vocabulary |
| DC-terms | Dublin core terms |
| DOLCE | Descriptive Ontology for Linguistic and Cognitive Engineering |
| DRR | Disaster Risk Reduction |
| DUL | DOLCE upper level ontology |
| EDxL | Emergency Data Exchange Language |
| EM-DAT | Emergency Events Database |
| EMO | Emergency Management Ontario |
| EU | European Union |
| EUDAT | European Union Collaborative data infrastructure project |
| ERO-M | Emergency Response Organization Ontology Model |
| EO | Earth Observation |
| FAIR | Findable Accessible, Interoperable and reusable |
| FEMA | Federal Emergency Management Agency |
| FOAF | Friend of a friend |
| GEMET | General Multilingual Environmental Thesaurus |
| GFO | General Formal Ontology |
| GNU | GNU's Not Unix |
| GUPRI | Globally Unique, Persistent and Resolvable Identifier |
| HXL | Humanitarian eXchange Language |
| HVR | Hazard Vulnerability and Risk Analysis |
| ICDRM | The Institute for Crisis, Disaster and Risk Management |
| icontact | International Contact Ontology |
| ICS | Incident Command System |
| IEEE | Institute of Electrical and Electronics Engineers |
| IRI | Internationalized Resource Identifier |
| ISyCri | Information Systems Interoperability in Crisis Situations |
| ISI | Institute for Scientific Information |
| KNMI | Koninklijk Nederlands Meteorologisch Instituut |
| LODE | An ontology for Linking Open Descriptions of Events |
| LOV | Linked Open Vocabularies |
| MIT | Massachusetts Institute of Technology |
| MOAC | the Management of a Crisis Vocabulary |
| MA_ont | Ontology for Media Resources |
| NIMS | National Incident Management System |
| OBO | Open Biological and Biomedical Ontology |
| ODPs | Ontology design patterns |
| ORFEUS | Observatories and Research Facilities for European Seismology |
| ONKI | Finnish Ontology Library Service ONKI |
| OWL | Web Ontology Language |
| OWL-S | Semantic Markup for Web Services |

| | |
|---|---|
| PESCaDO | Personalized Environmental Service Configuration and Delivery Orchestration |
| PRec (PRec.) | FAIR /metric/indicator |
| PROTON | PROTo ONtology |
| Prov-O | Provenance Ontology |
| QUOMOS | Quantities and Units of Measure Ontology Standard |
| RDF | Resource Description Framework |
| RDFS | Resource Description Framework Schema |
| Rec. | FAIR Recommendation |
| RVA | Risk and Vulnerability Analysis |
| SCOT | Social Semantic Cloud of Tags |
| SEM | Simple Event Model |
| SKOS | Simple Knowledge Organization System |
| SIIM | Spatial Image Information Mining |
| SIOC | Semantically-Interlinked Online Communities) Core Ontology |
| SLR | Systematic Literature Review |
| SSN ontology | Semantic Sensor Network ontology |
| SML | Situation Modeling Language |
| SUMO | Suggested Upper Merged Ontology |
| SWEET | Semantic Web for Earth and Environmental Terminology |
| SPARQL | SPARQL Protocol and RDF Query Language |
| SWRL | Semantic Web Rule Language |
| UFO | Unified foundational ontology |
| UNESCO-IOC | Intergovernmental Oceanographic Commission of UNESCO (IOC) |
| UNU-EHS | United Nations University, Institute for Environment and Human Security |
| UNDRR | United Nations Office for Disaster Risk Reduction |
| UNHCR | United Nations High Commissioner for Refugees |
| UNOCHA | United Nations Office for Disaster Risk Reduction |
| URI | Uniform Resource Identifier |
| W3C | World Wide Web Consortium |
| VUWIKI | Vulnerability Wiki |

## Appendix A

*Description of Reused Vocabularies*

Table A1 presents artefacts reused in the development of ontologies in the disaster management domain as illustrated in Figure 5.

**Table A1.** Reused Ontologies in the Disaster Management Domain.

| Vocabulary | Description |
|---|---|
| **Upper level ontologies** | |
| BFO | Foundational ontology that supports representation of information for purposes of retrieval, analysis and integration in a domain of interest |
| SWEET | Modular mid level ontology originally developed by NASA to capture concepts in Earth science and Environment domains |
| DOLCE | Foundational ontology developed within the EU WonderWeb project to capture the meanings for interoperation and consensus |
| SUMO | Upper ontology developed for with the original intention of representing human knowledge in computer information systems |
| ABC Ontology | upper level ontology that provides conceptual basis for representing existing metadata vocabularies and instances in web and digital repositories |
| UFO | Foundational ontology combining concepts from DOLCE, GFO and universals underlying ontoclean for representing conceptual modelling knowledge |
| CCO | Mid-level extension of BFO, developed with an intention of representing and integrating taxonomies of generic classes and relations in a domain of interest |
| PROTON | Upper level ontology that provides coverage for concepts for semantic annotation, indexing and retrieval |

**Table A1.** *Cont.*

| Vocabulary | Description |
|---|---|
| | STANDARD ONTOLOGIES |
| W3C Geospatial Vocabulary | ontology developed by W3C Geospatial Incubator Group (GeoXG) for representing geospatial knowledge |
| GeoSPARQL ontology | Defines SPARQL constructs for representing and querying geospatial data |
| Geonames ontology | Allows semantic description of geographical features defined in the geonames.org data base |
| BasicGeo (WGS84 long/lat) | Provides a namespace for geographically representing things using WGS84 as a reference datum. |
| SKOS | Defines a vocabulary for organising knowledge organization systems such as taxonomies, classification schemes, thesauri, etc |
| FOAF | Simple ontology for representing knowledge about persons, and their relationships |
| SIOC ontology | Describes information resources from online communities such as wikis, weblogs, etc. |
| OWL-Time | Standard OWL vocabulary for defining the temporal properties of resources |
| DC_terms | Specification for metadata elements developed and maintained by the Dublin Core Metadata Initiative |
| OWL-S | Semantic Markup for Web Services-OWL Vocabulary that describes web services |
| Roles and profiles ontology (WAI) | Extends the FOAF specification with concepts of roles and profiles |
| ALOCOM | Generic learning content model for learning objects and components |
| SSN ontology | Defines knowledge about sensors, their observations, and actuators. |
| icontact | Provides concepts and properties for representing street addresses, phone numbers and emails |
| OASIS (QUOMOS) ontology | Defines quantities, systems of measurement units, and base dimensions for use across multiple industries |
| SIIM ontology | Semantic framework for Linked Earth Observation Data that incorporates Topological relations in logic based reasoning |
| Location , places ontology | Represents knowledge about locations, such as administrative limits and coordinates based on the wgs84_pos vocubulary |
| Dbpedia ontology | Multilingual cross-domain ontology based on commonly used Wikipedia infoboxes |
| MA_ont | Defines a core vocabulary and a set of mappings between different metadata formats of media resources on the web |
| LODE ontology | Publishes descriptions of historical events as well as mapping of events in other vocabularies |
| Prov-O | Provides a vocabulary for recording information about entities, activities, and people involved in producing a thing in different contexts and systems . |
| SCOT | Defines concepts for expressing social tagging at a semantic level in a way a machine understands. |
| CiTO | An ontology that represents the nature or type of citations in a factual and rhetorical way |
| | DISASTER MANAGEMENT RELATED ONTOLOGIES |
| AGROVOC | Controlled vocabulary for organising knowledge in areas of food, nutrition, agriculture, fisheries, forestry and the environment for purposes of supporting data retrieval. |
| Geological Hazard Domain Ontology | Ontology for representing concepts in the Geological domain which is derived from the People's Republic of China for Geology and Mineral Industry Standards |
| MONITOR | Ontologies for risk management in the disaster domain developed under the MONITOR EU project |
| Community Domain Ontology (CDO) on Emergency Management | This ontology defines the basic vocabulary used in the emergency management domain |
| Disaster Management Core ontology | defines general concepts in the emergency field e.g., emergency events, risks, resources, goals, plans, etc. |
| EDXL | XML based language for emergency information sharing and data exchange across stakeholders via standardized messaging. |
| SoKNOS ontology | Core domain ontology for representing knowledge in the emergency management domain |
| Community-based ontology | An extensive ontology describing knowledge in a volcanic system |
| PESCaDO | Modular ontology for organising environmental data e.g., meteorological disasters |
| MOAC | Lightweight vocabulary for building consensus among practitioners on different "things" in crisis management. |
| HXL | Vocabulary that provides formal definition of the terminology used data sharing during humanitarian crisis. |

**Table A1.** *Cont.*

| Vocabulary | Description |
|---|---|
| Existing ODPs (e.g., Event, Quality, region ODPs) | Reusable modular ontologies published ontology design patterns at ODP wiki |
| Event-Model-F | A formal model of events intended to support interoperability in distributed event-based systems. It has been applied in the domain of emergency response. |
| SML. | A graphical language for situation modelling |
| CAP standard | Used for exchanging all-hazard alerts and public warnings |
| RDFG-names Rdf graphs | Describes the graph data model in ontologies |
| SEM | Simple Event Model (SEM) Ontology defines entities that describe an event |
| Weather [90] | Defines a formal domain model of the weather. Contains concepts such as pressure, temperature and visibility as events used for event extraction from news text |
| Disaster ontology | ontology listed by the Finish Library service ONKI/Finto for purposes of defining concepts on man-made and natural hazard to manage disaster situations. See http://onki.fi/en/browser/overview/disaster, accessed on 21 December 2020 |
| INFORMAL ONTOLOGIES (GLOSSARIES AND THESAURI) | |
| EM-DAT | Crisis/hazard related taxonomy developed by CRED for disaster preparedness and humanitarian actions at national and international level |
| FEMA glossary | Glossary that defines terms disaster preparation and management |
| Emergency and Crisis Communication Vocabulary | Provides a core terminology for emergency and crisis communication from Government Services Canada |
| EMO glossary | Produced by Ontario provincial government Working Group of EMO for purposes of emergency management. |
| NIMS glossary | Glossary from FEMA that enable stakeholders to work together to manage incidents |
| ICDRM glossary | Defines terminology for emergency management education and practice in the context of emergency response and recovery |
| ICS glossary | Provides common terminology for incident management. |
| UNU_EHS | Provides a core terminology for describing disaster risk, vulnerability and related concepts |
| UNDRR _glossary/ISDR | Common terminology by UNDRR that promotes understanding in the disaster risk reduction community |
| Australian emergency management terms thesaurus | Provides a list of terminologies and definitions for emergency management |
| GEMET | Multilingual thesauri developed by the European Environment Agency (EEA) to defines a common language/terminology for the environment |
| Disaster category classification | It is an initiative led by CRED and MunichRE that implemented as a common "Disaster Category Classification" based on EMDAT, Desinventar, NatCATservice and Sigma databases |
| Tsunamis (UNESCO-IOC ) | Provides a common definition of tsunami vocabulary for warning and mitigation among global intergovernmental coordination groups |
| Wildland fires | Defines terminology commonly used by NWCG working group for wildland fire and incident management |
| UNHCR glossary | Glossary with terminology and definitions of concepts used in humanitarian and crisis management |

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
