# Peer review of "Implementation of FAIR Principles for Ontologies in the Disaster Domain: A Systematic Literature Review"

_ijgi, doi:10.3390/ijgi10050324_

Round 1
Reviewer 1 Report
The article submitted for review is an extremely interesting review work that requires only slight corrections in terms of the scope and form of content presentation.
- The objectives given in the abstract do not entirely coincide with what is in the Introduction and in the following parts of the article - it should be harmonized
- Part 2 of the article - Related work - should be expanded both in terms of content - some concrete examples and literature. You can include, for example, work: Borowska-Stefańska, M., Kowalski, M., Turoboś, F., & Wiśniewski, S. (2019). Optimization patterns for the process of a planned evacuation in the event of a flood. Environmental Hazards, 18 (4), 335-360.
- Content included in some parts of the article, eg 3.1. 3.2. should be expanded based on the bullet points that are there. Please pay attention in particular to section 3.1. - where the goal is not exactly the same as before. For this, a short introduction should be written here, not just wording in the form of bullets
- In my opinion, Figure 2 - there is no need to put the data on the chart - it is enough to describe. The chart is of poor quality, takes up a lot of space and the amount of data it presents is negligible (the more that this data is already included in the content of the article and repeating them on the chart does not make sense). The same goes for Figures 3 (a and b) .
Author Response
The article submitted for review is an extremely interesting review work that requires only slight corrections in terms of the scope and form of content presentation.
reviewer Comment 1:
1. The objectives given in the abstract do not entirely coincide with what is in the Introduction and in the following parts of the article - it should be harmonized
Response 1
The abstract has been edited to capture content in the introduction, the problem and major findings of the paper
reviewer Comment 2:
2. Part 2 of the article - Related work - should be expanded both in terms of content - some concrete examples and literature. You can include, for example, work: Borowska-Stefańska, M., Kowalski, M., Turoboś, F., & Wiśniewski, S. (2019). Optimization patterns for the process of a planned evacuation in the event of a flood. Environmental Hazards, 18 (4), 335-360.
Response 2
we have improved related literature - in the literature review we have reviewed FAIR concepts, their relevance for disaster domain, and use-cases existing in the disaster management domain furthermore we have added a detailed description of perspectives that form the foundation for FAIR maturity assessment. Finally, we contextualize the FAIR concepts for semantic artefacts and provide disaster management examples where a few such metrics have been applied as well as their limitations. We also find the proposed citation above important for could guiding improvement of FAIR data management processes
reviewer Comment 3:
3. Content included in some parts of the article, eg 3.1. 3.2. should be expanded based on the bullet points that are there. Please pay attention in particular to section 3.1. - where the goal is not exactly the same as before. For this, a short introduction should be written here, not just wording in the form of bullets
Response 3
We have added content to explain the screening process for which the inclusion and exclusion criteria form a basis (line 162-169). short introduction sentences have been added on section 3.1 and 3.2 respectively.
reviewer Comment 4
4. In my opinion, Figure 2 - there is no need to put the data on the chart - it is enough to describe. The chart is of poor quality, takes up a lot of space and the amount of data it presents is negligible (the more that this data is already included in the content of the article and repeating them on the chart does not make sense). The same goes for Figures 3 (a and b).
Response 4
Figure 2 has been deleted. also, Data(percentages) has been removed from the chart in Figure three which now figures two. only text labels remain
Reviewer 2 Report
Please stablish a consecutive numbering of tables because tables 1 and 2 numbering are duplicated.
Figures has low quality for print. A vectorial format for them is needed.
Title in Figure 4 is not necessary. Increase space between the x-axis title and figure captions.

Author Response
Dear Reviewer,
Thank you for the review comments of our article "Implementation of FAIR Principles for Ontologies in the Disaster Domain: A Systematic Literature Review".
These comments have been extremely valuable to improving our paper, as well as stimulating new research agendas.
Based on these comments, we have made the necessary corrections to the manuscript.
Sincerely
Authors
Reviewer comment 1
The paper is positioned in the field of semantic computer analysis (eScience) and try to assess how well the disaster management science has been able to share data and outcomes because authors think that this is a way to determine the maturity of this science, that includes humanitarian crisis response, hazard vulnerability and risks, prevention, and mitigation of disasters. Authors try to identify effective barriers preventing the dissemination of data published in journals, websites, or books by means of various formal indicators, grouped under the acronym FAIR (Findable, Accessible, Interoperable, Reusable), using a semi-quantitative statistic. The most relevant problems highlighted in this article are that low numbers of URLs represent disaster management knowledge and 90% of the downloadable disaster management items do not conform to the principle of uniqueness and persistence. Only 1,4% of all retrieved ontologies are published in semantic repositories and 84,4 % are not published at all, basically, because there are no repositories dedicated to archiving disaster domain knowledge. This leads to a low capacity for Findability (1,7%) or Accessibility (5.77%), while Interoperability and Reusability are moderate (49% and 30,2 % respectively). With these results, the authors conclude that disaster data ntegration processes are incomplete. As Disaster Management Knowledge does not comply with the FAIR principles, the ability of using this previous knowledge during disaster management phases is limited. Perhaps could be most appropriate to have chosen a computer science or big data journal to publish this paper but may be interesting to readers of this journal because it notes a lack of capacity to share information that affects the practical efficacy for publications on disaster management issues.
Response 1
We totally agree with your assessment above.
Action: As such we have used some of the text above that has been incorporated in the Abstract of the paper as well as the introduction section of the paper. This in a way helps us align the abstract to the rest of the paper
Reviewer comment 2
For authors: The paper should express most clearly that implementation of the FAIR principles for ontologies can help to solve demands for interoperability, due to its specific ability to handle heterogeneous and diverse information in the Disaster Management Domain and to promote collaboration among various users and organizations.
Response 2
We totally agree with the comment above about the relevance of FAIR for ontologies.
Action: Therefore, we have harmonized our introduction section to include this comment.
Reviewer comment 3
Also, section1 should include information on the evolution and development of ontology concepts and taxonomies to evaluate the quality of different scientific domains
Response
We agree to the comment.
Action: section one has been edited to include Lines 34-46 which highlight the need for a coordinated approach to semantics to benefit ontology evolution and development
Reviewer Comment 4
Section 5 should include a comparison between the results obtained here with those of other domains (geospatial data, environment for instance), in order to check the degree of development and deficiencies in the disaster management field and explain why these differences.
Response
We agree to the comment.
Action: In the introductory part of section 5, we highlight that results from the Biomedical domain are more superior compared to the disaster management domain because of the requirement to publish in semantic repositories using best practice principles. However, only citations for reuse and accessibility were found in literature, while the findability was implied from the cited articles given that ontologies were drawn from the BIOPORTAL repository that is based on OBO foundry best practices.
Reviewer comment 5
A great distance exists between the accuracy of the text and the graphic quality of figures, which is generally low. A vectorial format for figures is needed.
Response 5
we agree with the comment.
Action: All figures were converted into pdf for better quality
Reviewer Comment 6 Title in Figure 4 is not necessary. Increase space between the x-axis title and figure captions
Response: 6
we agree to the comment
Action: the x-axis for images was increased and the title in figure 4, now figure 3 was deleted
Reviewer comment 7
Origin of tables and figures must be indicated if they come from a source outside the work Response 7
We agree with the comment.
Action: only table 2, was extracted from an external source-.. these sources are Mentioned. However, other tables emerge from data.
Reviewer comment 8
Please establish a consecutive numbering of tables because tables 1 and 2 numberings are duplicated Some tables are too much longer and should be subdivided according to some categorical criteria.
Response 8:
we agree with this comment.
Action 1) Continuous numbering has been affected for all tables.
Action 2) All tables have been categorized with headings. for instance, table 4 represents ontologies in emergency response and while 5 in preparedness and early warning, hazard vulnerability and risk etc. similarly table 12 and 13 ontologies have been categorized according to the ontology hierarchy
Reviewer 3 Report
I believe every non-ironic or non-curmudgeon reader of this test would agree that FAIR principles are important for data dissemination. They certainly merit discussion. This article goes a long way in studying if FAIR-Principles are implemented in the disaster _response_ domain. It fails to engage with the most important question, esp. given the unsurprising findings how FAIR-Principles matter. A major part of problem is organisational, enhanced by a very uneven treatment of the issues and material. Structurally, the author need to go beyond citing FAIR and explain it; provide an example from the disaster _response_ domain, articulate its strengths and weaknesses.
Just charging ahead with analysis would be fine if the foundation for the analysis was strong. But it is not and the paper meander.
I find the paper can be rewritten around the points from the Discussion of Results section effectively. It’s an important paper that can and should get a clear focus and organization.
More specific points
The literature review approach needs to be explained in some detail.
The relevance of ontologies from FAIR-Principles needs great attention.
Applications of FAIR should be presented in their relevance for the findings
Complex ontologies do not make complex understanding, maybe just a great deal of detail.
The maturity model and FAIR maturity evaluation should be detailed to frame the analysis and methods
Is ‘elicit’ the right verb? Draw out speaks more to the research analysis you present and avoid suggesting some kind of elusive interpretation.
Author Response
Dear Reviewer,
Thank you for the review comments of our article "Implementation of FAIR Principles for Ontologies in the Disaster Domain: A Systematic Literature Review".
These comments have been extremely valuable to improving our paper, as well as stimulating new research agendas.
Based on these comments, we have made the necessary corrections to the manuscript.
Sincerely
Authors
Comments and Suggestions for Authors
Reviewer comment 1
I believe every non-ironic or non-curmudgeon reader of this test would agree that FAIR principles are important for data dissemination. They certainly merit discussion. This article goes a long way in studying if FAIR-Principles are implemented in the disaster _response_ domain. It fails to engage with the most important question, esp. given the unsurprising findings how FAIR-Principles matter. A major part of problem is organisational, enhanced by a very uneven treatment of the issues and material. Structurally, the author need to go beyond citing FAIR and explain it; provide an example from the disaster _response_ domain, articulate its strengths and weaknesses.
Response 1
We agree to the comment:
Action: we understood the comment as requiring us to expand related work and include concrete FAIR use cases in the domain. we have edited section 2 to provide applications of FAIR in the disaster management domain. examples are drawn from management of data for climate change impacts(floods) and seismic hazards. More detail is provided in response 4 below
reviewer comment 2
Just charging ahead with analysis would be fine if the foundation for the analysis was strong. But it is not and the paper meander.
Response 2:
We agree to the comment: we understood the comment as requiring us to elaborate on foundation and perspectives for maturity assessment in the methodology section:
Action: see response 8 for the discussion about the framing of methods based on the maturity model
Reviewer comment 3
I find the paper can be rewritten around the points from the Discussion of Results section effectively. It’s an important paper that can and should get a clear focus and organization.
Response 3
We agree with the comment.
Action: the abstract, sections 1, 2 and 3 of Paper have been re worked to align the work with section 4 and 5
Reviewer comments 4
More specific points
The literature review approach needs to be explained in some detail.
Response 4
We agree to the comment
Action: we have improved related literature - in the literature review we have reviewed FAIR concepts, its relevance for disaster domain, and use-cases existing in the disaster management domain. Furthermore, we have added a detailed description of perspectives that form the foundation for FAIR maturity assessment. Finally, we contextualize the FAIR concepts for semantic artefacts and provide disaster management examples where a few such metrics have been applied as well as their limitations.
Reviewer comments 5
The relevance of ontologies from FAIR-Principles needs great attention.
Response 5
We agree to the comment
Action: We have edited the abstract, lines 40-44 of the introduction, 63-65 related work to provide insights into the relevance of ontologies from fair principles
Reviewer comment 6
Applications of FAIR should be presented in their relevance for the findings
Response 6
We agree to the comment
Action: We edit the abstract and section 1 to include the relevance of FAIR to ontologies and related applications such as evolution of ontologies, data interoperability. Similarly, in section 5 lines 448-462 we introduce applications of FAIR in light of the findings such as API development, evolution of knowledge. we further explain these more in the implications for FAIR to research and industry.
Reviewer comments 7
Complex ontologies do not make complex understanding, maybe just a great deal of detail.
Response 7
We agree to the comment. We understood comment as requiring us to add detail to complex ontology description.
Action: Additional detail has been added to text in section 4 and 5 where we felt detail was needed. However, all ontologies are categorized, described and their respective abbreviations are explained in the appendix.
Reviewer comment 8
The maturity model and FAIR maturity evaluation should be detailed to frame the analysis and methods
Response 8
We agree to this comment. we also find this comment extremely useful in grounding our methodology
Action: The discussion on the maturity models and existing perspectives to measurements are highlighted in section 2 (related work). According to the cited document, the draft guidelines are not meant to be normative; they intend to provide guidelines to inform assessment approaches but leave the way it is implemented to the evaluator. From these guidelines, we finally chose one perspective "Measuring pass-or-fail” to assess fairness of ontologies in the disaster management domain in section 3. Therefore, beyond these guidelines, we also make a few assumptions that are custom to our study in light of limitations presented. For instance, the Authors of the guidelines don’t consider perspectives at a metric level such as the community view, the repository view and individual ontology view. similarly, weighting of indicators doesn’t exist yet in the semantic ecosystem.
Reviewer comment 9
Is ‘elicit’ the right verb? Draw out speaks more to the research analysis you present and avoid suggesting some kind of elusive interpretation
Response 9
We agree to the comment
Action: the word "elicit" has been replaced with appropriate terms
Round 2
Reviewer 3 Report
The reorganization provides a clearer focus. I think just a few small items remain to "clean up".
- compliance is more of a legal term. Given FAIR principles are just principles, I'd suggest using non-adherence rather than non-compliance throughout the text.
- In the abstract the connection between the maturity of sharing and FAIR principles is not clearly explained.
- The first paragraph on p. 2 needs revision. It is not clear what "its" refers to in the fourth sentence, starting on line 41.
- Same paragraph, the sentence of line 47: what are the "coordinated recomendations"?
- Same paragraph, last line. Please cite important works from the "limited literature". This will be helpful for readers.
- p. 4, line 144: please explain more clearly and cite literature for the method "forward and backward" directions are not associated with snowball sampling.
- p. 20, line 556ff: given the limited uptake of FAIR principles you describe in the paper and your point on p. 19 about the need for future work examining trade-offs, it seems this last sentence of the paper could point to the importance of better understanding the tradeoffs, rather than evaluate them using a clear scheme, which however doesn't yet exist.
Author Response
Dear Reviewer,
Thank you for the review comments of our article "Implementation of FAIR Principles for Ontologies in the Disaster Domain: A Systematic Literature Review".
These comments have been extremely valuable to improving our paper.
Based on these comments, we have made the necessary corrections to the manuscript.
Sincerely
Authors
Comments and Suggestions for Authors
The reorganization provides a clearer focus. I think just a few small items remain to "clean up".
Reviewer Comment 1
compliance is more of a legal term. Given FAIR principles are just principles; I'd suggest using non-adherence rather than non-compliance throughout the text.
Response 1
We agree with the comment
ACTION: We have replaced the term non-compliance with “adherence” in our text.
Reviewer Comment 2
In the abstract, the connection between the maturity of sharing and FAIR principles is not clearly explained.
Response 2
We agree with the comment
ACTION: We have added line 5-6 in the abstract which explains the Role of FAIR principles in Data sharing. We have also edited line 8 and 20 to be able to bring out the connection.
Reviewer Comment 3
The first paragraph on p. 2 needs revision. It is not clear what "its" refers to in the fourth sentence, starting on line 41. Same paragraph, the sentence of line 47: what are the "coordinated recommendations"?
Response 3
We agree to the comment
ACTION: We have revised paragraph one on page 2 i.e line 39-50 to make it clearer. – the coordinated recommendations meant best practices for publishing and sharing semantic artefacts,
Reviewer Comment 4
Same paragraph, last line. Please cite important works from the "limited literature". This will be helpful for readers.
Response 4.
We agree to the comment
ACTION: We have added a citation from
Albris, K., Lauta, K.C. & Raju, E. Disaster Knowledge Gaps: Exploring the Interface Between Science and Policy for Disaster Risk Reduction in Europe. Int J Disaster Risk Sci 11, 1–12 (2020). https://doi.org/10.1007/s13753-020-00250-5
This Highlights the disaster knowledge gaps in the domain. A key knowledge gap identified is a terminological gap that hinders communication with the policy domain and the public.
We also added a citation from Lui et al 2013 that reviews ontologies for representing emergency management knowledge.
Reviewer Comment 5
- 4, line 144: please explain more clearly and cite literature for the method "forward and backward" directions are not associated with snowball sampling.
Response 5
We agree to the comment
ACTION: We have clarified on the use of snowballing in the SLR. In this SLR, we extracted additional papers from reference lists of selected papers. This is known as the backward snowballing
We provide a citation to explain the snowballing method in
Wohlin, C. Guidelines for snowballing in systematic literature studies and a replication in software engineering. Proceedings of the 18th international conference on evaluation and assessment in software engineering, 2014, pp. 1–10
Reviewer Comment 6
- 20, line 556ff: given the limited uptake of FAIR principles you describe in the paper and your point on p. 19 about the need for future work examining trade-offs, it seems this last sentence of the paper could point to the importance of better understanding the tradeoffs, rather than evaluate them using a clear scheme, which however doesn't yet exist.
Response 6
We agree to the comment
ACTION: We have edited line 556 to point to the importance of a better understanding of Tradeoffs in FAIR principles